# Genome-Wide CpG Island Methylation Profiles of Cutaneous Skin with and without HPV Infection

**DOI:** 10.3390/ijms20194822

**Published:** 2019-09-28

**Authors:** Laith N. AL-Eitan, Mansour A. Alghamdi, Amneh H. Tarkhan, Firas A. Al-Qarqaz

**Affiliations:** 1Department of Applied Biological Sciences, Jordan University of Science and Technology, Irbid 22110, Jordan; amneht92@gmail.com; 2Department of Biotechnology and Genetic Engineering, Jordan University of Science and Technology, Irbid 22110, Jordan; 3Department of Human Anatomy, College of Medicine, King Khalid University, Abha 61421, Saudi Arabia; m.alghamdi@kku.edu.sa; 4Department of Internal Medicine, Jordan University of Science and Technology, Irbid 22110, Jordan; fqarqaz@just.edu.jo; 5Division of Dermatology, Department of Internal Medicine, King Abdullah University Hospital Jordan University of Science and Technology, Irbid 22110, Jordan

**Keywords:** HPV, warts, DNA methylation, CpG, epigenetics

## Abstract

HPV infection is one of the most commonly transmitted diseases among the global population. While it can be asymptomatic, non-genital HPV infection often gives rise to cutaneous warts, which are benign growths arising from the epidermal layer of the skin. This study aimed to produce a global analysis of the ways in which cutaneous wart formation affected the CpG island methylome. The Infinium MethylationEPIC BeadChip microarray was utilized in order to quantitatively interrogate CpG island methylation in genomic DNA extracted from 24 paired wart and normal skin samples. Differential methylation analysis was carried out by means of assigning a combined rank score using RnBeads. The 1000 top-ranking CpG islands were then subject to Locus Overlap Analysis (LOLA) for enrichment of genomic ranges, while signaling pathway analysis was carried out on the top 100 differentially methylated CpG islands. Differential methylation analysis illustrated that the most differentially methylated CpG islands in warts lay within the *ITGB5*, *DTNB*, *RBFOX3*, *SLC6A9*, and *C2orf27A* genes. In addition, the most enriched genomic region sets in warts were Sheffield’s tissue-clustered DNase hypersensitive sites, ENCODE’s segmentation and transcription factor binding sites, codex sites, and the epigenome sites from cistrome. Lastly, signaling pathway analysis showed that the *GRB2*, *GNB1*, *NTRK1*, *AXIN1*, and *SKI* genes were the most common regulators of the genes associated with the top 100 most differentially methylated CpG islands in warts. Our study shows that HPV-induced cutaneous warts have a clear CpG island methylation profile that sets them apart from normal skin. Such a finding could account for the temporary nature of warts and the capacity for individuals to undergo clinical remission.

## 1. Introduction

The human papillomaviruses (HPV) comprise a diverse family of DNA viruses that infect a number of avian, mammalian, and reptilian species [1]. Over 170 types of HPV have been fully characterized, and these types can either infect the cutaneous integumentary system or the mucosal linings of the human body [2]. Additionally, the aforementioned HPV types can be classified as low- or high-risk depending on their carcinogenic potential [3]. In immunocompetent individuals, cutaneous HPV infection often manifests as benign tumors known as warts, which are common among the general population and children in particular [4]. Nonetheless, reports indicate that the presence of cutaneous HPV types is implicated in the development of keratinocyte cancers such as squamous cell carcinomas [5]. In fact, HPV infection has been associated with modulated oral, oropharyngeal, and laryngeal squamous cell carcinoma risk and prognoses in various populations [6,7,8,9]. Furthermore, reports have shown that such HPV-associated cancers are associated with changes in epigenetic processes, namely DNA methylation [3,10,11].

DNA methylation (DNA-m) is an epigenetic phenomenon that most commonly involves the transfer of methyl groups to the cytosines of CpG dinucleotide sites, which is a process that can be reversed and/or inherited by subsequent generations [12]. Maintaining certain patterns of DNA-m is essential for normal cell- and tissue-specific expression, and differences in DNA-m contribute to the natural variation found between discrete human populations [13]. The vast majority (70–80%) of CpG sites in humans are methylated, rendering them transcriptionally silent [14]. However, unmethylated CpG sites are mostly found in 1000-bp-long, GC-rich clusters known as CpG islands, the latter of which are associated with the majority of annotated gene promoters in the human genome and are strongly correlated with the initiation of transcription [15]. Aberrant CpG island methylation patterns have been reported for a number of human diseases, including many types of cancer as well as HPV infection [16,17,18].

Owing to the well-established link between certain HPV types and cervical cancer, the focus of epigenetic HPV research has been on those types that infect the cervical epithelia [19]. HPV-mediated DNA methylation has been implicated in the etiology of cervical cancer as well as keratinocyte cancer [18,20,21,22]. Similarly, the methylation status of HPV DNA itself has been explored, with studies showing that methylation levels differed between HPV types and that methylation at certain CpG sites could act as a cervical precancer biomarker [23,24]. In contrast to HPV-associated cancers, warts have been the subject of considerably less attention and research on the molecular level. However, the transient nature of warts strongly suggests that there are HPV-induced epigenetic changes in the methylation patterns of CpG islands, the latter of which are mostly found in the promoter regions of genes [25]. To the best of the authors’ knowledge, no previously published study has addressed or compared CpG methylation patterns in benign HPV-induced cutaneous warts compared to healthy skin. Therefore, the aim of the present study was to investigate the levels of methylation at CpG sites in cutaneous skin with and without HPV infection. 

## 2. Results

### 2.1. Sample Clustering

Samples showed an expected clustering based on all methylation values of the top 1000 most variable loci (Figure 1). Samples sharing similar methylation patterns or phenotypes tended to cluster together. Additionally, the dataset was subjected to principal component analysis (PCA) in order to inspect for a strong signal in the methylation values of the samples (Figure 2). PCA confirmed that the difference between warts (W) and normal skin (NS) dominates the analysis. 

### 2.2. Differential Methylation of CpG Islands

A total of 866,895 probes included in the MethylationEPIC array were processed with the RnBeads Bioconductor package. We excluded probes that overlapped with single nucleotide polymorphisms (SNPs) (*n* = 17,371), probes of highest impurity (*n* = 2310), and probes of the specified contexts (*n* = 2980). A total of 844,234 probes were retained, of which a total of 26,550 CpG islands passed quality control and pre-processing. The assessment of methylation value distributions of the CpG islands for W and NS showed notable differences in β values (Figure 3A). Also, more methylation was observed in the CpG islands compared to other CpG contents (open sea, shelf, and shore) (Figure 3B). The list of DM CpG islands in W was limited to the top-ranking 1000 CpG islands using the “combined rank score”. Using this scoring method, 550 CpG islands were found to be hypomethylated and 450 CpG islands were hypermethylated in W compared to NS, with a mean β difference ≥ 0.046 and ≤−0.046, and a *p*-value ≤ 0.006 (adjusted *p*-value ≤ 0.076) (Figure 4A). Of the 550 hypomethylated CpG islands, the β difference ranged from −0.046 to −0.300, and of the 450 hypermethylated CpG islands, the β difference ranged from 0.046 to 0.414. The log_2_ of the quotient in methylation between W and NS had a maximum value of 1.888 and minimum value of −1.138 (Figure 4B). The top 100 CpG islands with the lowest “combined rank score” are presented in Appendix A with the associated gene names.

### 2.3. Hypermethylated and Hypomethylated CpG Islands

LOLA enrichment analysis of the top 1000 hypermethylated and top 1000 hypomethylated CpG islands are shown in Appendix A, respectively.

### 2.4. Signaling Networks

Analysis of the genes located in the top 100 DM CpG islands showed that five genes were found to be common regulators with a minimum of 12 connectivities. These genes are *GRB2*, *GNB1*, *NTRK1*, *AXIN1,* and *SKI* (Figure 5). 

### 2.5. Validation of the Top Five Genes Associated with Differentially Methylated CpG Islands

Differentially methylated CpG islands were ranked using the “combined ranking score” (Appendix A). Genes of interest included the five top-ranked genes (*ITGB5*, *DTNB*, *RBFOX3*, *SLC6A9* and *C2orf27A*), which were selected for preliminary validation. The expression and methylation levels of these genes in warts compared to normal skin has been measured and can be seen in Table 1. The *ITGB5* and *DTNB* genes have been found to be significantly differentially methylated in warts. Moreover, *ITGB5* is up-regulated in warts compared to normal skin.

## 3. Discussion

Since CpG islands are mostly found within promoter regions, their methylation statuses often determine whether a particular gene is transcribed or silenced [15]. Promoter-associated CpG islands are almost always unmethylated except during periods of cell development and differentiation [26]. In HPV-infected keratinocytes, apoptosis is inhibited and cell division is stimulated, leading to a number of transient cellular changes that culminates in the formation of a wart [27]. The impermanence of warts points towards the HPV-induced modulation of methylation patterns in warts compared to normal skin [3]. The present study provided a genome-wide characterization of the CpG island methylation profiles of 24 paired wart and normal skin samples from 12 Jordanian-Arab males.

The top-ranking CpG islands in cutaneous warts were found within the integrin subunit beta 5 (*ITGB5*), dystrobrevin beta (*DTNB*), RNA binding protein, fox-1 homolog 3 (*RBFOX3*), solute carrier family 6 member 9 (*SLC6A9*), and uncharacterized protein C2orf27 (*C2orf27A*) genes. These genes exhibited different methylation and expression patterns in warts compared to normal skin (Table 1). In a porcine model, the *ITGB5* gene was reported to be a significant adhesion molecule of mucosal epithelial signaling in the response to *Escherichia coli*, while, in humans, *ITGB5* variants were associated with airway hyperresponsiveness [28,29]. The *ITGB5* gene was also found to be a key gene of the most highly enriched pathways in Het-1A cells expressing L2, the minor capsid protein of HPV [30]. Although the exact function of the *DTNB* gene is still unclear, its involvement in neural differentiation processes has been suggested [31]. Similarly, the *RBFOX3* gene has been associated with neuronal differentiation as well as neurological disorders like Rolandic epilepsy and sleep latency [32,33]. In contrast, the *SLC6A9* gene was found to be upregulated in normal human keratinocytes treated with epidermal growth factor receptor (EGFR) inhibitors [34]. Little is known about the *C2orf27A* gene in terms of its function, but it has been previously reported to be associated with the Ebola virus disease [35].

Among the top 1000 hypermethylated CpG islands in cutaneous warts, the most enriched genomic region set comprised the tissue-clustered DNase hypersensitive sites from Sheffield, namely primitive epithelial, skin, epithelial, and normal human epidermal keratinocyte (NHEK) sites. Similarly, the most enriched genomic region sets among the top 1000 hypomethylated CpG islands were the tissue-clustered DNase hypersensitive sites from Sheffield (normal human dermal fibroblast (NHDF), hematopoietic, and fibroblast epithelial sites), the ENCODE segmentation sites, the ENCODE transcription factor binding sites, the codex sites, and the epigenome sites from cistrome. In untransformed human fibroblasts, DNase hypersensitive regions were found to be associated with CpG sites in which methylation negatively corresponds with gene expression [36].

Signaling pathway analysis of the top 100 DM CpG islands revealed that the most common regulators were the growth factor receptor-bound protein 2 (*GRB2*), guanine nucleotide-binding protein G(I)/G(S)/G(T) subunit beta-1 (*GNB1*), neurotrophic tyrosine kinase receptor type 1(*NTRK1*), axin 1 (*AXIN1*)*,* and ski proto-oncogene (*SKI*) genes. The *GRB2* gene is critical in the downstream transduction of *EGFR*, and the aberrant expression of the latter has been implicated in several inflammatory skin diseases [37,38]. *GRB2* was also found to play a major role in the life cycle of high-risk HPV types as well as associated carcinogenesis [39,40]. De novo mutations in the *GNB1* gene have been associated with cutaneous mastocytosis and neurodevelopmental delays, while mutations in the *NTRK1* gene were found to cause anhidrosis and palmar hyperkeratosis [41,42]. In contrast, *AXIN1* expression is often dysregulated in various types of cancer cells [43,44,45,46,47]. On a similar note, the *SKI* oncogene has been linked to several HPV-associated cancers such as esophageal squamous cell carcinoma and cervical cancer [48,49,50].

## 4. Methods 

### 4.1. Subjects 

Ethical approval to conduct the present study was given by the Institutional Review Board (IRB) at Jordan University of Science and Technology (JUST), with the ethical approval number of 2/105/2017. After obtaining written informed consent, twelve Arab males with common warts were recruited from the general population. All of the warts investigated in the current study were clinically identified as *Verruca vulgaris*, the latter of which is associated with HPV types 1, 2, 3, 4, 5, 7, 27, 29, and 57 [51,52]. Paired tissue samples of warts and adjacent normal skin were obtained from each participant via shave biopsy, a procedure which excises only the epidermal layer of the skin. In terms of anatomical localization, 20 samples were obtained from the hands, 2 from the feet, and 2 from the forehead. The 24 samples were then stored at −20 °C until further processing.

### 4.2. DNA Extraction and the Infinium MethylationEPIC BeadChip

Genomic DNA was extracted by means of a QIAamp DNA Mini Kit (Qiagen, Hilden, Germany), and optional RNase A digestion was performed. The BioTek PowerWave XS2 Spectrophotometer (BioTek Instruments, Inc., Winuski, VT, USA) was employed in order to determine DNA purity, while agarose gel electrophoresis was carried out in order to examine DNA integrity. For the wart samples only, a second and heavier band was exhibited, pointing towards the presence of the circular dsDNA of HPV. Samples that met the standards for purity and integrity were then shipped on dry ice to the Melbourne node of the Australian Genome Research Facility (AGRF). At the AGRF, the quality of the samples was assessed by the QuantiFluor^®^ dsDNA System (Promega, Madison, WI, USA) and resolution was evaluated via 0.8% agarose gel electrophoresis. The 24 samples were then normalized to approximately 500 ng of DNA in 45 μL and bisulfite converted with Zymo EZ DNA Methylation kit (Zymo Research, Owen, CA, USA). Afterwards, the Infinium MethylationEPIC BeadChip microarray (Illumina, San Diego, CA, USA) was used to carry out a genome-wide analysis of the methylation patterns of over 850,000 CpG sites.

### 4.3. Data Analysis and Processing

Data analysis was carried out on Illumina’s GenomeStudio v2011.1 with Methylation module 1.9.0 software, using the default Illumina settings and MethylationEPIC_v-1-0_B3 manifest file. All samples were above 850,000 detected CpG (*p-*value = 0.01). A computational R package (RnBeads) was adapted to process and analyze the raw intensity data from the methylation chip (IDAT files) [53]. Quality control, preprocessing, batch effects adjustment, and normalization were carried out on all probes and samples according to the RnBeads package pipeline. 

### 4.4. Differential Methylation Analysis

Differential methylation analysis was performed on CpG islands, at which the mean of the mean β (mean.mean β) values of all the tested CpG sites were computed. The distribution of the CpG sites per and across CpG islands can be seen in Figure 6A,B, respectively. Differential methylation (DM) for each CpG island was calculated using three measures: the mean.mean β difference between W and NS, the log_2_ of the mean quotient in β means across all CpG sites in a CpG island, and the adjusted combined *p*-value of all CpG sites in the CpG islands using a limma statistical test [54]. The Benjamini-Hochberg (B-H) procedure was used to correct for multiple testing. Moreover, these three measures were used to give each CpG island a rank, and the combined rank was computed as the maximum (=worst) rank among the three ranks. CpG islands which exhibit more DM will have a smaller combined rank [53]. CpG islands were sorted from smallest to largest using the “combined ranking score”, and the top-ranking 1000 CpG islands were selected for further analysis.

### 4.5. Locus Overlap Analysis

Locus Overlap Analysis (LOLA) for enrichment of genomic ranges was conducted for the best 1000 CpG islands [55]. The following LOLA reference databases collections were used in the analysis: the Cistrome database from Cistrome (cistrome_cistrome), the Epigenome databases from Cistrome (cistrome_epigenome), the Codex database (CODEX), chromatin segmentation states from ENCODE (encode_segmentation), transcription factor binding sites from ENCODE (ENCODE_TFBS), tissue-clustered DNase hypersensitive sites from Sheffield et al. (2013) (Sheffield_dnase), and a collection of UCSC feature tables (UCSC_features). The LOLA tool uses Fischer’s exact test and produces a ranked list of significantly enriched region sets.

### 4.6. Signaling Pathway Analysis

Signaling Network Open Resource (SIGNOR) 2.0 investigates the causal relationships between biological entities, and it was used to explore the signaling network of the genes associated with the top 100 most DM CpG islands [56]. The type of relation was selected to include “all” interactions with a relaxed layout and score of “0.0”. 

### 4.7. Validation of the Genes Associated with DM CpG Islands

Five genes which have been identified to be associated with the top-ranked DM CpG islands were subjected to further validation by examining their expression and methylation levels. These genes were validated by reviewing the results of two studies previously carried out in our lab. Using the same samples involved in the present study, one study investigated DNA methylation levels while the other measured expression levels in warts compared to normal skin [57].

## 5. Conclusions

In the present study, genomic DNA was extracted from paired wart and normal skin tissue samples and subjected to methylation sequencing. Differential methylation analysis revealed certain genes to be significantly hyper- and hypomethylated in warts compared to normal skin. Our findings indicate that host CpG island methylation is an integral part of the HPV infection and wart formation processes. The reversibility of DNA-m could potentially explain wart remission and clearance. To the best of the authors’ knowledge, this is the first study to compare and contrast the CpG methylation status between HPV-induced warts and adjacent normal skin. Future research is necessary to identify the functional and clinical relevance of the significantly associated CpG sites.

## Figures and Tables

**Figure 1 ijms-20-04822-f001:**
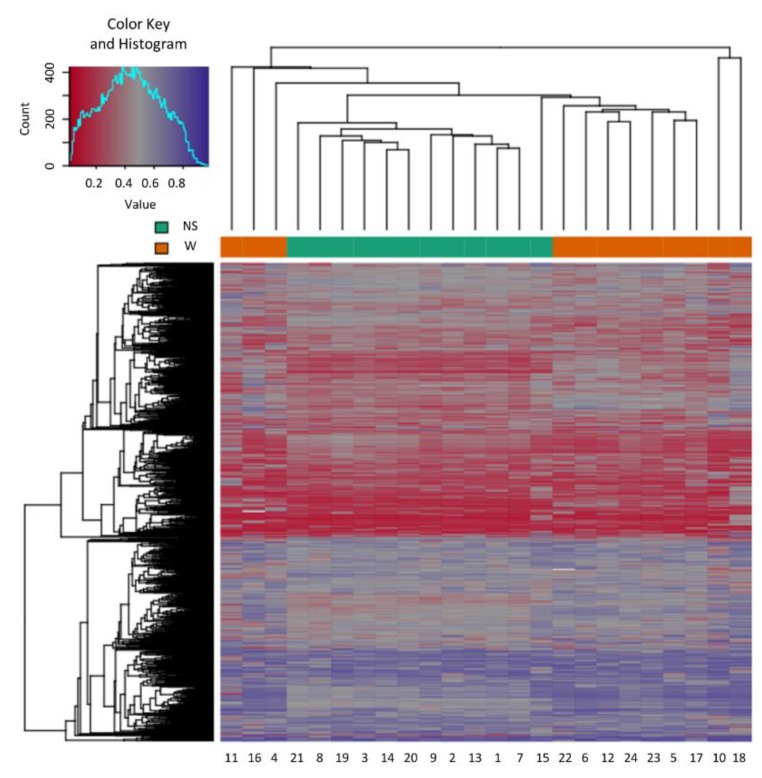
Heatmap showing the hierarchal clustering of the 1000 most variable loci with the highest variance across the wart (W) and normal skin (NS) samples. Clustering incorporated average linkage and Manhattan distance. The numbers at the bottom horizontal axis (1 to 24) stand for the patient samples.

**Figure 2 ijms-20-04822-f002:**
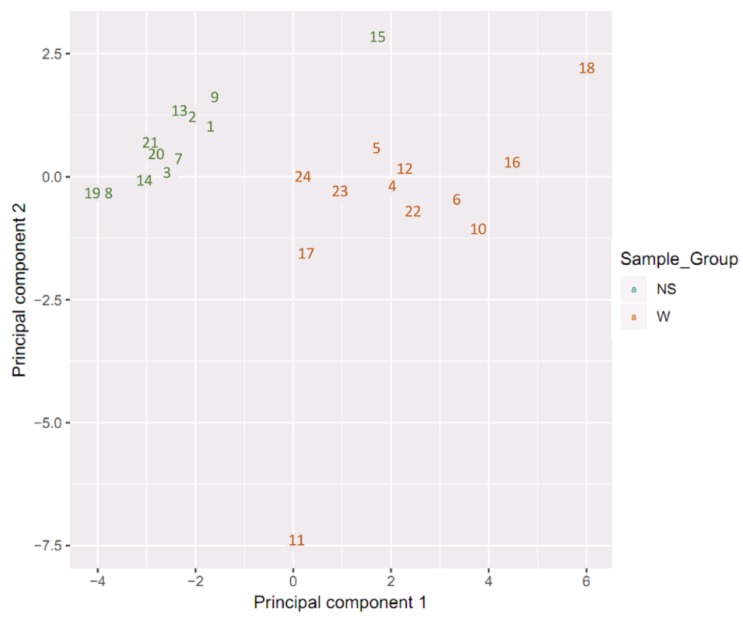
Scatter plot showing normal skin (NS) and wart (W) samples’ coordinates on the first and second principal components. Discrimination between the NS and W samples is shown as in indicated by the color capture.

**Figure 3 ijms-20-04822-f003:**
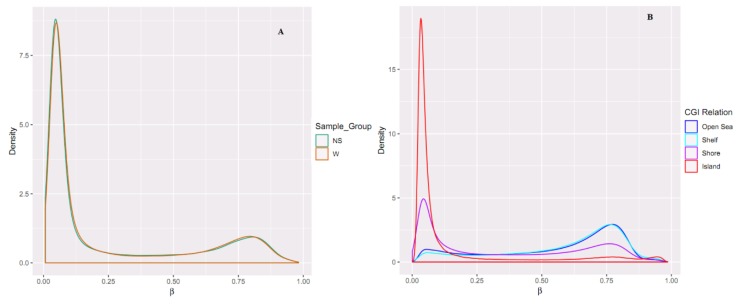
Comparing the density distributions of methylation levels (β) in (**A**) W and NS and in (**B**) the CpG content. In (**B**), it can be seen that the island relation has more β value density than other CpG content.

**Figure 4 ijms-20-04822-f004:**
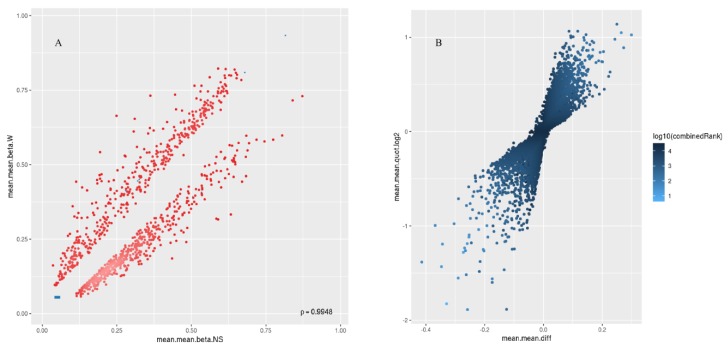
Scatterplot (**A**) and volcano plot (**B**) analysis for the top 1000 most differentially methylated CpG islands. In (**A**), the mean of mean methylation levels (beta value) for normal skin across all sites in a region (mean.mean.beta.NS) is on the x axis, and the mean of mean methylation levels (beta value) for warts across all sites in a region (mean.mean.beta.W) is on the y axis. Methylation beta values ranged from 0 (unmethylated) to 1 (fully methylated). In (**B**), the figure illustrates the overall extent of hypermethylation (>0) and hypomethylation (<0) of the top 1000 most differentially methylated CpG islands. Differential methylation was quantified by log_2_ of the mean quotient in means across all sites in a region (mean.mean.quot.log_2_) and the mean difference in means across all sites in a region (mean.mean.diff) between W and NS.

**Figure 5 ijms-20-04822-f005:**
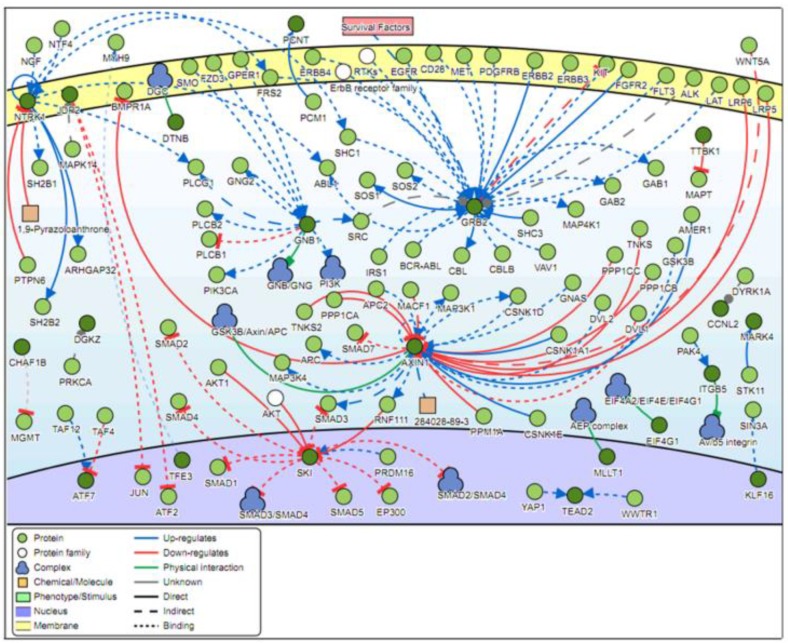
Pathway signaling network generated from the genes located within the top 100 most differentially methylated CpG islands. Five genes (*GRB2, GNB1, NTRK1, AXIN1*, and *SKI*) have a minimum of 12 connectivities.

**Figure 6 ijms-20-04822-f006:**
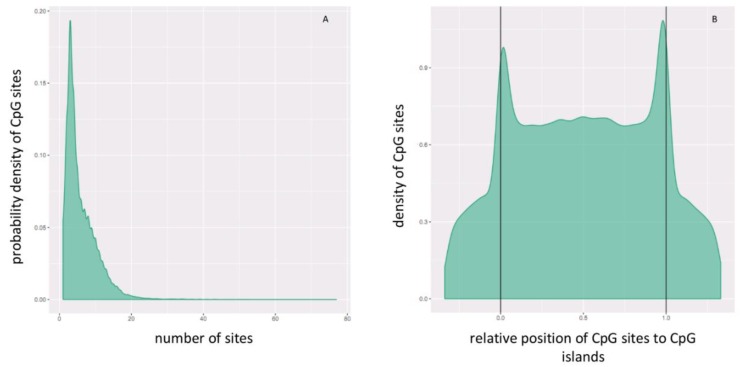
CgG site distribution (**A**) per and (**B**) across CpG islands. In panel B, the relative coordinates of 0 and 1 correspond to the start and end coordinates of each CpG island. Coordinates smaller than 0 and greater than 1 denote flanking regions normalized by region length. CpG sites are more dense at the start and end of CpG islands.

**Table 1 ijms-20-04822-t001:** Methylation and expression levels of the genes associated with the five top-ranked CpG islands in warts (W) compared to normal skin (NS).

	Methylation Levels				Expression Levels	
Gene	Mean.mean β value (NS)	Mean.mean β value (W)	Mean.mean β value (W-NS)	*p*-value	logFC	*p*-value
*ITGB5*	0.386	0.428	−0.042	0.0002	0.418	0.0060
*DTNB*	0.608	0.604	0.004	0.0138	−0.029	0.8569
*RBFOX3*	0.493	0.473	0.021	0.0543	n/a	n/a
*SLC6A9*	0.497	0.499	−0.002	0.0602	0.212	0.3504
*C2orf27A*	0.600	0.598	0.002	0.2827	n/a	n/a

Mean.mean β value = mean of mean methylation levels across all sites in a region; logFC = log fold change.

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
