# Peer review of "Genome-Wide CpG Island Methylation Profiles of Cutaneous Skin with and without HPV Infection"

_ijms, 2019, doi:10.3390/ijms20194822_

Round 1

Reviewer 1 Report

Al-Eitan et al.characterized the methylation profiles of biopsy samples from warts (W) and adjacent areas of healthy skin (NS) using the Infinium MethylationEPIC BeadChip microarray. This is an interesting topic, but unfortunately some of the basic informations are missing from the manuscript, I think. In Methods, the anatomical localization of the warts and the adjacent skin area studied is missing. The histology of the biopsies is also lacking. There is no background information regarding the presence or absence of human papillomavirus (HPV) DNA in the biopsy samples, either. In addition, the type of HPV, if present, would be an important detail to clarify because it may affect the interaction of the viral genomes with the host cells. Some the results are described in a cryptic manner; in Results, 2.2., the authors mentioned that a total of 262,550 CpG islands passed quality control and pre-processing. There is no data, however, regarding the number of CpG islands analysed in the two sample types (W and NS) or in individual samples. In table 1., the "top 100" differentially methylated CpG islands are compiled, and the genes associated with 5 of them which are  highly methylated in warts compared to normal skin are delt with in Discussion. I think that control bisulfite sequencing of full-length CpG          islands in case of the "top 5" CpG islands would add interesting details to this work. On page 4, Results, 2.4., Signaling Networks, 5 genes claimed to be common regulators of subsets from the top 100 differentially methylated CpG islands are listed and the sentence refers to Figure 13. Unfortunately, however, the protein products of these genes (GRB2, GNB1, NTRK1, AXIN1, SKI) are not shown on Figure 13. In Discussion, page 5, line 41, promoter-associated CpG islands are mentioned; the reference supporting this sentence (ref.30, Jeziorska et al., 2017) deals, however, with intragenic CpG islands. The athors may wish to discuss the significance of CpG islands which are less methylated in warts compared to normal skin.

I recommend a major revision of this manuscript before publication.

Author Response

I would like to extend my deepest thanks to yourself and to the reviewers for their constructive comments and suggestions with regard to the manuscript titled “Genome-wide CpG island methylation profiles of cutaneous skin with and without HPV infection”. I am pleased to submit the revised version of the paper that includes a point-by-point response to the reviewer comments:

Comments by Reviewer 1

Al-Eitan et al.characterized the methylation profiles of biopsy samples from warts (W) and adjacent areas of healthy skin (NS) using the Infinium MethylationEPIC BeadChip microarray. This is an interesting topic, but unfortunately some of the basic informations are missing from the manuscript, I think.

1. In Methods, the anatomical localization of the warts and the adjacent skin area studied is missing.

Added the following sentence to sub-section 4.1 of the Methods section:

In terms of anatomical localization, 20 samples were obtained from the hands, 2 from the feet, and 2 from the forehead.

2. The histology of the biopsies is also lacking.

Histological analysis was not conducted. However, shave biopsies only excise the epidermal layer of the skin, leaving the dermal layer intact. This information was added to sub-section 4.1 of the Methods section.

3. There is no background information regarding the presence or absence of human papillomavirus (HPV) DNA in the biopsy samples, either.

Gel electrophoresis was carried out for both the NS and W samples. Only the W samples exhibited an extra band near the wells of the gel, indicating the presence of circular HPV DNA. Moreover, electropherogram analysis revealed an extra peak in the W samples only, which points towards the presence of another species, i.e. HPV dsDNA. This information was added to sub-section 4.1 of the Methods section.

4. In addition, the type of HPV, if present, would be an important detail to clarify because it may affect the interaction of the viral genomes with the host cells.

The Infinium MethylationEPIC BeadChip kit is human-specific, and the viral genome is too small (8 kb compared to the 3235 Mb of the human genome) to pose any contamination issues. In the current study, HPV typing was not carried out. Nonetheless, all of the warts excised in the current study were of the Verruca vulgaris type. It has already been established that the HPV genotypes most commonly associated with Verruca vulgaris are HPV 1, 2, 3, 4, 5, 7, 27, 29, and 57 [1,2]. This information was added to sub-section 4.1 of the Methods section.

5. Some the results are described in a cryptic manner; in Results, 2.2., the authors mentioned that a total of 262,550 CpG islands passed quality control and pre-processing. There is no data, however, regarding the number of CpG islands analysed in the two sample types (W and NS) or in individual samples.

Added and clarified data in sub-section 2.2 regarding the number of CpG islands analyzed:

A total of 866895 probes included in the MethylationEPIC array were processed with the RnBeads Bioconductor package. We excluded probes that overlapped with single nucleotide polymorphisms (SNPs) (n=17371), probes of highest impurity (n=2310), and probes of the specified contexts (n=2980). A total of 844234 probes were retained, of which a total of 26550 CpG islands passed quality control and pre-processing.

6. In table 1., the "top 100" differentially methylated CpG islands are compiled, and the genes associated with 5 of them which are highly methylated in warts compared to normal skin are delt with in Discussion. I think that control bisulfite sequencing of full-length CpG islands in case of the "top 5" CpG islands would add interesting details to this work.

Because of budgetary and technical constraints, bisulfate sequencing is presently unfeasible at our department. However, as components of two other studies which are currently under review, we have carried out RNA-sequencing and methylation analysis on the same samples investigated in the current study. As a form of validation, we contrasted the gene expression and methylation levels for the five genes associated with the top-ranked DM CpG islands in warts compared to normal skin (Table 1).

7. On page 4, Results, 2.4., Signaling Networks, 5 genes claimed to be common regulators of subsets from the top 100 differentially methylated CpG islands are listed and the sentence refers to Figure 13. Unfortunately, however, the protein products of these genes (GRB2, GNB1, NTRK1, AXIN1, SKI) are not shown on Figure 13.

Corrected this issue by replacing Figure 13 with the correct figure (Fig 5) for this analysis.

8. In Discussion, page 5, line 41, promoter-associated CpG islands are mentioned; the reference supporting this sentence (ref.30, Jeziorska et al., 2017) deals, however, with intragenic CpG islands.

The reference discusses promoter-associated CpG islands at several instances, namely:

a. While CGIs associated with promoters nearly always remain unmethylated, many of the ∼9,000 CGIs lying within gene bodies become methylated during development and differentiation.
b. In particular, while most promoter CGIs remain unmethylated, ∼9,000 CGIs within gene bodies (intragenic) are more likely to become methylated.
c. The majority of promoter (∼95%) and alternative promoter CGIs (∼85%) are unmethylated (≤45% methylation).

9. The athors may wish to discuss the significance of CpG islands which are less methylated in warts compared to normal skin.

Unfortunately, very little information is available regarding the genes associated with the most hypomethylated CpG islands in warts (ZN407, TEX11, CHMP1A, HECW1, PNMA6A).

Reviewer 2 Report

The study “Genome-wide CpG island methylation profiles of cutaneous skin with and without HPV infection” by Al-Eitan et al. examines the levels of methylation at CpG sites in cutaneous skin with and without HPV infection. Although in contrast to HPV-associated cervical cancers, warts (benign growths arising from the epidermal layer of the skin) have been the subject of considerably less attention and research on the molecular level. To the best of the authors’ knowledge, no previously published study has addressed or compared CpG methylation patterns in benign HPV-induced cutaneous warts compared to healthy skin indicating that the data depicted in the manuscript could provide some novelty in the field of science.

\\ MAJOR concerns:

Figure 1 and 2 are somehow lost. The authors started the result section from figure 3. Please, correct. Figures are designed in a way that it takes too much space for the massage they are depicting. Figure 4 and 5 are depicting the same massage, the authors should select only one for the front figure. I did not even understand what significant data is represented by figure 8… Table 1 is too huge to be represented as main table, please make it as supplement. Figures 9 – 12 should be much more representative. The authors depict: “LOLA enrichment analysis of the top 1000 hypermethylated and top 1000 hypomethylated CpG islands are shown in Figures 9 and 10 and Figures 11 and 12, respectively.” and refer it to fig 9-12. I suggest to move these figures to supplement. Signaling Networks are represented in very unclear way. The authors show figure 13, where I can see many genes depicted; however, the authors depicted 5 genes which are not signed in the figure. Furthermore, it is not clear, how these genes where selected. The methylome data must be deposited in open access database otherwise it is not possible to calrify/verify or compare the data by other scientists. Methylome of five most distinguished genes should be validated by additional methods (for instance qPCR). The validity of the present data remain very uncertain….

Author Response

I would like to extend my deepest thanks to yourself and to the reviewers for their constructive comments and suggestions with regard to the manuscript titled “Genome-wide CpG island methylation profiles of cutaneous skin with and without HPV infection”. I am pleased to submit the revised version of the paper that includes a point-by-point response to the reviewer comments:

Comments by Reviewer 2

The study “Genome-wide CpG island methylation profiles of cutaneous skin with and without HPV infection” by Al-Eitan et al. examines the levels of methylation at CpG sites in cutaneous skin with and without HPV infection. Although in contrast to HPV-associated cervical cancers, warts (benign growths arising from the epidermal layer of the skin) have been the subject of considerably less attention and research on the molecular level. To the best of the authors’ knowledge, no previously published study has addressed or compared CpG methylation patterns in benign HPV-induced cutaneous warts compared to healthy skin indicating that the data depicted in the manuscript could provide some novelty in the field of science.

Figure 1 and 2 are somehow lost. The authors started the result section from figure 3. Please, correct.

Corrected.

Figures are designed in a way that it takes too much space for the massage they are depicting.

Moved Figures 9, 10, 11, and 12 to the supplementary section and combined others to reduce space.

Figure 4 and 5 are depicting the same massage, the authors should select only one for the front figure.

Selected Figure 5 for use in the paper and removed Figure 4.

I did not even understand what significant data is represented by figure 8…

Figure 8 illustrated the extent of hypermethylation (>0) and hypomethylation (<0) of the top 1000 most differentially methylated CpG islands. Nonetheless, Figures 7 and 8 were combined in order to increase flow in line with the text.

Table 1 is too huge to be represented as main table, please make it as supplement.

Moved Table 1 to the supplementary section.

Figures 9 – 12 should be much more representative. The authors depict: “LOLA enrichment analysis of the top 1000 hypermethylated and top 1000 hypomethylated CpG islands are shown in Figures 9 and 10 and Figures 11 and 12, respectively.” and refer it to fig 9-12. I suggest to move these figures to supplement.

Moved Figures 9, 10, 11, and 12 to the supplementary section.

Signaling Networks are represented in very unclear way. The authors show figure 13, where I can see many genes depicted; however, the authors depicted 5 genes which are not signed in the figure. Furthermore, it is not clear, how these genes where selected.

Corrected this issue by replacing Figure 13 with the correct figure (Fig 5) for this analysis.

The methylome data must be deposited in open access database otherwise it is not possible to calrify/verify or compare the data by other scientists.

The raw methylome data is available. However, at this stage, we cannot upload the raw data to an open access database because it is still being processed and used to generate other manuscripts which have been submitted for publication. We will upload the full dataset as soon as the peer review process is complete. Additionally, the complete processed methylation data for the CpG islands is available as a CSV file and will be available as a supplementary file.

Methylome of five most distinguished genes should be validated by additional methods (for instance qPCR). The validity of the present data remain very uncertain….

In two other studies which are currently undergoing review, we have carried out RNA-sequencing and methylation analysis on the same samples investigated in the current study. As a form of validation, we contrasted the gene expression and methylation levels for the five genes associated with the top-ranked DM CpG islands in warts compared to normal skin (Table 1).

Please do not hesitate to contact me if you need any additional information. I highly look forward to hearing from you.

Yours sincerely,

Dr. Laith N. AL-Eitan, MSc, PhD
Associate Professor of Human Genetics and Pharmacogenetics 

Department of Biotechnology & Genetic Engineering  
Faculty of Science and Arts
Jordan University of Science and Technology,
P.O.Box 3030, Irbid 22110, JORDAN
Email: [email protected]
Tel.: +962-2-7201000 ext.: 23464 

Reviewer 3 Report

The manuscript titled "Genome-wide CpG island methylation profiles of cutaneous skin with and without HPV infection", authored by Al-Eitan is scientifically sound and appropriate work to be published in IJMS. However, the presentation of the work with proper background is something missing.

Broadly, Authors successfully established their vision and the significance and novelty of this work. The background is also presented nice but the presentation of the results, conclusion to this work and proper summary is missing. When I read the whole manuscript the continuity of the story is missing. Authors may consider minimising the figures or including a graphical abstract or summary figure to make it simple.

Precisely, I do not have much corrections to suggest.

A reference for the statement, Line 37 in background "The transient nature of warts strongly suggest that there are HPV-induced epigenetic changes.. " needs a reference Spelling and grammar, formatting check is recommended, For example, space between number 2, and Results title (Line 78),

Author Response

I would like to extend my deepest thanks to yourself and to the reviewers for their constructive comments and suggestions with regard to the manuscript titled “Genome-wide CpG island methylation profiles of cutaneous skin with and without HPV infection”. I am pleased to submit the revised version of the paper that includes a point-by-point response to the reviewer comments:

Comments by Reviewer 3

The manuscript titled "Genome-wide CpG island methylation profiles of cutaneous skin with and without HPV infection", authored by Al-Eitan is scientifically sound and appropriate work to be published in IJMS. However, the presentation of the work with proper background is something missing.

Broadly, Authors successfully established their vision and the significance and novelty of this work. The background is also presented nice but the presentation of the results, conclusion to this work and proper summary is missing. When I read the whole manuscript the continuity of the story is missing.

Modified the Conclusions section to include a concise summary of this work.

Authors may consider minimising the figures or including a graphical abstract or summary figure to make it simple.

Added a graphical abstract in lieu of certain figures, the latter of which were moved to the supplementary section.

A reference for the statement, Line 37 in background "The transient nature of warts strongly suggest that there are HPV-induced epigenetic changes.. " needs a reference

Added reference.

Spelling and grammar, formatting check is recommended, For example, space between number 2, and Results title (Line 78), 

Manuscript was revised by a native English speaker in order to enhance its flow and level of scientific communication.

Please do not hesitate to contact me if you need any additional information. I highly look forward to hearing from you.

Yours sincerely,

Dr. Laith N. AL-Eitan, MSc, PhD
Associate Professor of Human Genetics and Pharmacogenetics 

Department of Biotechnology & Genetic Engineering  
Faculty of Science and Arts
Jordan University of Science and Technology,
P.O.Box 3030, Irbid 22110, JORDAN
Email: [email protected]
Tel.: +962-2-7201000 ext.: 23464 

Round 2

Reviewer 1 Report

The Authors correctly addressed the points I made. I recommend the revised manuscript for publication.

Author Response

Dear Mr. Cheng,

All minor revisions mentioned by the reviews have been appropriately addressed as can be seen from the below point-by-point response.

Minor Revisions

The Reviewers reached a consensus that the manuscript can be accepted in its present, revised form. However, for final acceptance of the manuscript, the quality of figures and figure captions should be further improved.

There are some small, but very disturbing details which need correction. For example, in Fig. 1, the text "Color Key and Histogram" is not placed correctly,

Corrected.

The meaning of numbers at the bottom, horizontal axis is missing and should be indicated at the axis.

Added the following sentence to the legend of Fig. 1: The numbers at the bottom horizontal axis (1 to 24) stand for the patient samples.

The green and orange numbers in Fig. 2 are too small.

Enlarged green and orange numbers.

In Fig. 4 texts such as "mean.mean.beta" on both axes should be replaced with a meaningful text or if used as abbreviation, it should be explained clearly in the figure caption.

Added an explanation of all abbreviations in Figure 4’s legend.

In Fig. 6, axis labels should be correctly re.written, such as "Probability density", "Number of sites", etc. Abbreviations as axis titles decrease readability of the manuscript.

Corrected.

Similar changes should be introduced for figs in the supplementary. 

Added abbreviations where necessary for the supplementary figures.

I would also advise to add 1-2 sentences of explanations to figure captions, where applicable.

Added pertinent sentences to all figure captions.

Please do not hesitate to contact me if you need any additional information.

Yours sincerely,

Dr. Laith N. AL-Eitan, MSc, PhD
Associate Professor of Human Genetics and Pharmacogenetics 

Department of Biotechnology & Genetic Engineering  
Faculty of Science and Arts
Jordan University of Science and Technology,
P.O.Box 3030, Irbid 22110, JORDAN
Email: [email protected]
Tel.: +962-2-7201000 ext.: 23464 

Reviewer 2 Report

The authors took a look and replied to all reviewer's suggestions.

Author Response

(The authors gave the same response as above.)
